# The Role of Fenugreek in the Management of Type 2 Diabetes

**DOI:** 10.3390/ijms25136987

**Published:** 2024-06-26

**Authors:** Melina Haxhiraj, Kenneth White, Cassandra Terry

**Affiliations:** Diabetes Interest Group, The Centre for Health and Life Sciences Research, London Metropolitan University, London N7 8DB, UK

**Keywords:** type 2 diabetes, fenugreek, 4 hydroxy isoleucine, therapeutics, natural compounds, microbiome

## Abstract

The number of people diagnosed with type 2 diabetes is on the increase worldwide. Of growing concern, the prevalence of type 2 diabetes in children and youths is increasing rapidly and mirrors the increasing burden of childhood obesity. There are many risk factors associated with the condition; some are due to lifestyle, but many are beyond our control, such as genetics. There is an urgent need to develop better therapeutics for the prevention and management of this complex condition since current medications often cause unwanted side effects, and poorly managed diabetes can result in the onset of related comorbidities. Naturally derived compounds have gained momentum for preventing and managing several complex conditions, including type 2 diabetes. Here, we provide an update on the benefits and limitations of fenugreek and its components as a therapeutic for type 2 diabetes, including its bioavailability and interaction with the microbiome.

## 1. Introduction

Type 2 Diabetes Mellitus (T2DM) is a complex metabolic disorder resulting from an interplay of both modifiable and non-modifiable risk factors. Modifiable risk factors include diet, exercise and socioeconomic disparities [1,2,3,4], whilst non-modifiable risk factors include age, gender, ethnicity and genetics [5,6,7,8,9,10]. Other risk factors include having gestational diabetes [11], long-term use of certain medications, depression, alcohol intake and smoking [12]. T2DM is a complex condition, and understanding the condition at the molecular level is crucial for developing and testing new and improved therapeutics for better disease management and prevention.

Treating diabetes and diabetes-related complications has a significant financial burden on healthcare systems worldwide, and these increases have paralleled rises in obesity [1]. Increases in youth-onset T2DM have risen disproportionately in ethnic minority groups [13], highlighting the need for urgent healthcare interventions.

Diabetes mellitus is attributed to dysfunctions in both the secretion and physiological activity of insulin marked by elevated plasma glucose levels, termed hyperglycemia [14]. Pathophysiologically, T2DM is characterized by insulin resistance and impaired insulin signaling pathways, leading to an inadequate cellular response to insulin. Insulin resistance reduces glucose uptake and pancreatic beta cell dysfunction, leading to persistent hyperglycemia [15]. This prolonged hyperglycemic state contributes to diverse complications, encompassing microvascular issues such as neuropathy and nephropathy, along with macrovascular complications, predominantly affecting the cardiovascular system [16]. Prediabetes is defined as those with an impaired glucose tolerance, with higher than normal blood glucose levels, who are at a high risk of developing diabetes. In contrast, type 1 diabetes occurs due to autoimmune-mediated destruction of pancreatic beta cells, culminating in reduced insulin synthesis [14].

### Current Methods for the Management of Type 2 Diabetes

It has been shown that T2DM is preventable and reversible through healthy lifestyle changes [17]. For those individuals diagnosed with T2DM and unable to reverse the condition through lifestyle changes, managing their blood glucose levels through prescribed medication is required. Commonly prescribed medications include metformin, sulfonylureas, thiazolidinediones, dipeptidyl peptidase-4 inhibitors, glucagon-like peptide-1 (GLP-1) receptor agonists, sodium-glucose cotransporter-2 (SGLT2) inhibitors and insulin therapy. These medications primarily aim to enhance insulin sensitivity, decrease hepatic glucose production, stimulate insulin secretion, or facilitate glucose uptake by peripheral tissues [18]. However, many of the currently prescribed medications lose efficacy over time, are expensive, have side effects and can interfere with other medications [18]. Reported side effects include gastrointestinal issues [19], higher cardiovascular disease risk and weight gain [20], acute pancreatitis [21], nausea, vomiting and diarrhea [22], increased incidence of urinary tract infections and diabetic ketoacidosis [23]. Inadequate management of diabetes can result in poorly controlled blood sugar levels, predisposing individuals to a myriad of complications and a higher risk of incurable conditions such as Alzheimer’s disease [16]. There is, therefore, a need for better preventative treatments and therapeutics to manage glucose levels better.

There have been several studies demonstrating that fenugreek (*Trigonella foenum-graecum*) can improve glycemic control in individuals with diabetes by reducing blood glucose levels and improving insulin sensitivity in obese and overweight patients with T2DM [24,25] and non-insulin-dependent diabetics [26,27]. Fenugreek supplementation has also shown benefits in lowering cholesterol levels and reducing the risk of cardiovascular diseases [28], thus reducing the risk of T2DM onset. Hence, here we discuss the benefits and limitations of fenugreek for T2DM management.

## 2. The Role of Natural Compounds in Diabetes Management

### 2.1. Promising Naturally Derived Compounds for Diabetes Management

Natural compounds are bioactive substances derived from natural sources such as plants, animals, minerals and microorganisms. Natural products have been trusted as traditional medicines across the world for centuries, with over 80% of the world’s population relying on them for treating various conditions [29]. Their exploitation for development into therapeutics has gained momentum due to their natural abundance, efficacy and being inexpensive and accessible. Potential limitations of their widespread use by the pharmaceutical industry include challenges in identifying the bioactive compound and extracting and producing it in large quantities [30]. In the context of treating T2DM, numerous natural compounds have shown promise [31,32,33,34,35,36,37,38,39,40,41,42,43]. These compounds can vary widely in their chemical structure and biological activity and exert their effects through various mechanisms, such as improving insulin sensitivity, enhancing glucose uptake, reducing intestinal glucose absorption, and modulating glucose metabolism, showing promising results in vitro and in vivo [31]. Metformin, the most commonly prescribed medication for those with T2DM, was developed from a naturally derived compound, galegine, extracted from *Galega officinalis* (or ‘Goat’s rue) [44].

In T2DM research, the majority of natural products tested are polyphenols [31,32,39]. Polyphenolic compounds are 500–4000 Daltons, have several phenolic hydroxyl groups on aromatic rings in their structure and are water-soluble. Numerous studies show that the flavonoid resveratrol (3,5,4’-trihydroxystilbene, found in grapes and blueberries) can improve glycemic control [45,46]; however, these studies include small sample sizes and short intervention periods (8 weeks) limiting an assessment of their long-term use in large, diverse populations. In addition, many of these studies have been conducted using resveratrol in combination with other natural compounds, thus limiting an assessment of the efficacy of resveratrol alone [47]. The antidiabetic effects of resveratrol are thought to be attributed to its ability to reduce body weight, increase glucose metabolism, increase insulin sensitivity and inhibit phosphodiesterase thus leading to an increase in insulin secretion [47].

Bitter melon (*Momordica charantia*) has also shown promising pre-clinical and clinical antidiabetic effects due to its high polyphenolic content, reducing fasting blood glucose levels and improving glucose tolerance [48]. Polyphenols in black tea have been shown to target enzymes such as α-glucosidase, involved in carbohydrate metabolism and blood glucose regulation [49]. In silico studies, several more polyphenolic compounds have been identified as potential inhibitors of alpha-glucosidase and alpha-amylase, suggesting a promising direction for future drug discovery [50]. Other promising compounds for T2DM management include alkaloids such as berberine [33], saponins such as soyasaponins and ginsenosides [34], fatty acids such as Omega-3 [35], polysaccharides [36], terpenoids [37,38,39,40,41,42] and organosulfur compounds [43].

### 2.2. Fenugreek as a Promising Natural Treatment for the Management of Type 2 Diabetes

Fenugreek (*Trigonella foenum-graecum*) is a member of the Fabaceae family, native to the Mediterranean region, Western Asia and Southern Europe. Fenugreek seeds and leaves are composed of macronutrients, vitamins, minerals and are rich in bioactive compounds like alkaloids (trigonelline), flavonoids (quercetin, luteolin), saponins and steroidal sapogenins. Additionally, fenugreek seeds are palatable and nutritionally dense, containing carbohydrates, proteins, lipids and volatile oils [51], making them useful for digestive health and as a source of plant-based nutrition. Their composition is listed in Table 1.

Fenugreek seeds possess several pharmacological properties such as anti-inflammatory, antioxidant, hypoglycemic, hypocholesterolemic and antiulcerogenic effects [52]. In traditional medicine, fenugreek has been used for centuries to treat a wide range of conditions, including digestive disorders and skin inflammation [53]. Steroidal saponins such as diosgenin and gitogenin have been shown to be the principal active constituents responsible for its medicinal properties [54]. Fenugreek has also been shown to inhibit the growth of cancer cells, induce apoptosis [55], and influence gut microbiota homeostasis in chickens [56], highlighting its additional health benefits.

There have been numerous studies showing that fenugreek has antidiabetic effects in vitro [57,58,59,60,61,62,63] in animal studies [52,59,64,65,66,67,68,69,70,71,72,73,74,75] and in trials with obese and diabetic patients [76,77,78,79,80,81]. The hypoglycemic effects of fenugreek seeds in animal studies have been summarized previously [64] and shown to be largely attributed to their high soluble dietary fiber content (galactomannan), trigonelline, diosgenin, 4-hydroxyisoleucine and flavone C-glycosides. Improved glucose tolerance was observed in streptozotocin-induced diabetic rats administered fenugreek (soluble dietary fiber fraction at 0.5 g per kilogram body weight) for 28 days, in addition to increased liver glycogen content and reduced serum glucose levels [72]. Similar results were observed in another study with diabetic rats given 0.44, 0.87 and 1.74 g per kilogram per day of fenugreek extract for 6 weeks [73]. They showed decreased glycated hemoglobin levels and decreased fasting blood glucose in comparison to untreated diabetic rats. Similarly, diabetic rats orally administered fenugreek extract at 0.1, 0.25 and 0.5 g per kilogram body weight for 14 days had reported decreased serum glucose and increased insulin levels [74]. Fenugreek seed extracts administered at lower concentrations of 50, 100 and 200 milligrams per kilogram to diabetic rats for 6 weeks also reported reduced blood glucose levels compared to non-treated diabetic controls [75].

Since fenugreek has shown promising antidiabetic effects in vitro and in numerous animal studies, several clinical trials have recently been undertaken to assess its antidiabetic effects in humans. Neelakatan et al. analyzed the results of 10 separate clinical trials assessing the efficacy of fenugreek seeds on T2DM [77]. Reduced glucose levels were observed in patients after daily doses of 1–100 g of fenugreek seed (median 25 g) over 10–84 days (median 30 days). The results revealed that fenugreek notably improved glycemic control and enhanced insulin sensitivity in individuals with T2DM; however, the studies were inconclusive due to the wide range of doses, from 1 to 100 g of seed or seed extract and participants having differing diabetes status. A meta-analysis of randomly controlled clinical trials conducted with T2DM patients showed that across the studies, fenugreek seeds reduced HbA1c levels (glycated hemoglobin where glucose adheres to red blood cells); however, these studies are difficult to compare due to heterogeneity in study design and differing doses administered [78].

Another study involving 18 T2DM participants showed that fenugreek seeds reduced their fasting blood sugar and triglycerides when given 10 g of powdered fenugreek seeds for 8 weeks [79]. While promising, 18 individuals is a small test group and, therefore, would need to be repeated with a larger cohort.

More recently, Kim et al. [80] performed a systematic review and meta-analysis to assess the effectiveness of fenugreek in individuals with T2DM and prediabetes. In total, they analyzed 10 studies involving 706 participants and found that fenugreek significantly reduced fasting blood glucose levels, HbA1c and 2-h plasma glucose level (2-hPG) in the intervention groups compared to the untreated control groups. In addition, fenugreek appeared to improve lipid levels (triglyceride, total cholesterol and high-density lipoprotein), which are known risk factors for T2DM. Fenugreek did not significantly influence body mass index, reflecting similar observations in C57BL/6J mice [59]. This meta-analysis highlighted the limitations and heterogeneity in study design that influenced the results collected. There was only one study focused on prediabetes [81]; hence, the data are biased towards the efficacy of fenugreek in T2DM, not prediabetes. Non-standardized methods include differences in fenugreek extraction, preparation, study duration and dosage and number of participants. Further studies are needed using standardized methods with a larger number of participants over longer time periods to assess the effectiveness of fenugreek in T2DM. Also, further studies assessing the effects of fenugreek in prediabetes are urgently needed as a potential early intervention therapeutic to prevent the onset of T2DM.

### 2.3. Toxicity and Adverse Effects

Since fenugreek is consumed through diet, it is considered safe to ingest; however, this is dependent on the doses consumed, and different studies report conflicting results. Animal experiments have been used to assess the toxicity of fenugreek seed extract [65]. Doses below 1000 milligrams per kilogram weight showed no toxicity in Wistar rats or their offspring after 90 days of repeated doses. Another study with leaf glycosidic fenugreek extracts administered to mice (at 0.2–1 g per kilogram for up to 7 days) showed minimal adverse effects [82]. However, due to its estrogenic activity, fenugreek seeds are not recommended to be consumed during pregnancy. Studies with rabbits showed that fenugreek affected fetal development when pregnant rabbits were fed 30% fenugreek seeds for three months [83].

In humans, the results are conflicting [80]. No reports of severe adverse reactions such as renal or hepatic toxicity were reported across ten independent studies, although mild gastrointestinal side effects were reported in some individuals, albeit short-lived. Another study used the ‘ToxRTool’ to assess the reliability of published toxicology studies for fenugreek seeds [84]. From the initial 436 results, only 17 studies were deemed ‘reliable without restrictions’ and showed that standardized extraction of fenugreek was non-toxic, with only a few reports of gastrointestinal side effects. A study analyzing sera from 29 different patients who have IgE specific to peanuts and certain legumes revealed that sensitization to fenugreek is a consequence of cross-reactivity in patients with peanut allergy [85]; hence, it should not be given to prediabetics or T2DM patients with peanut or legume allergies. Further studies are clearly needed. These should include larger cohorts of participants from diverse backgrounds monitored over a long period of time, focusing on dose responses. However, initial studies suggest fenugreek may be a safe and effective therapeutic for those with T2DM.

**Table 1 ijms-25-06987-t001:** The composition of fenugreek shows the percentage present in leaves and seeds. Adapted from [86].

Constituent Type	Constituent	Fenugreek Seeds (Approximate %)	Fenugreek Leaves (Approximate %)
Macronutrients	Proteins	20–30%	4–6%
	Dietary Fibre	25–30%	3–5%
	Carbohydrates	40–60%	5–10%
	Fats	5–10%	<1%
Vitamins	B Vitamins (e.g., B6, B12)	0.3–0.6%	0.1–0.2%
	Vitamin K	Trace amounts	0.2–0.4%
	Vitamin A	Trace amounts	0.02–0.05%
	Vitamin C	0.1–0.3%	0.3–0.7%
	Folate (Folic Acid)	0.02–0.05%	0.01–0.03%
	Vitamin E	Trace amounts	Trace amounts
Minerals	Iron	1–2%	2–4%
	Calcium	0.2–0.5%	1–3%
	Magnesium	0.1–0.3%	0.2–0.5%
	Phosphorus	0.3–0.6%	0.2–0.4%
	Potassium	0.4–0.8%	0.5–1%
	Sodium	<0.1%	<0.1%
	Zinc	0.01–0.03%	0.01–0.03%
Bioactive Compounds	Saponins	2–6%	<1%
	Alkaloids	<1%	Trace amounts
	4-hydroxyisoleucine	1–2%	Trace amounts
	Flavonoids	0.5–1%	0.2–0.5%
	Phenolic Acids	0.1–0.3%	0.1–0.2%
	Coumarins	Trace amounts	Trace amounts
	Lecithin	0.1–0.3%	Trace amounts
Other Components	Water Content	5–10%	75–85%
	Essential Oils	Trace to 1%	Trace amounts
	Mucilage	1–3%	<1%
	Choline	0.02–0.05%	Trace amounts

### 2.4. Bioactive Components of Fenugreek

Several studies have attempted to extract and purify individual components of fenugreek to test them in relation to diabetes management. Diosgenin, a saponin (steroidal glycoalkaloid), constitutes 3–6% of the seeds and 0.6–0.8% of the leaves. Streptozotocin-induced diabetic rats fed a single dose (5 or 10 milligrams per kilogram per body weight) of diosgenin for 30 days showed increased regeneration of pancreatic beta cells and insulin granules [66].

Trigonelline, categorized as an alkaloid, comprises 0.1–0.3% and 0.6–1.2% of fenugreek seeds and leaves, respectively. Trigonelline can reduce blood glucose concentrations in rats induced with diabetes through a high-fat and low-streptozotocin diet [67]. Its ability to increase insulin sensitivity is due to its ability to improve the insulin signaling pathway by increasing insulin receptor autophosphorylation (IR-PH) and the translocation of effector molecules pT308-Akt, and glucose transporter 4 to the cell surface, thus increasing glucose uptake [68].

Galactomannan (a polysaccharide composed of galactose side groups and a mannose backbone) is a carbohydrate prevalent in fenugreek seeds, constituting 40–50% of their composition and 4–6% in leaves (Table 1). Purified galactomannan reduced fasting hyperglycemia and improved serum and hepatic lipid profiles in control and streptozotocin-induced diabetic adult male Wistar rats [69]. Clinical trials involving 24 patients with T2DM who were given 8 or 16 g of an oral galactomannan derivative showed it to be safe under the conditions tested [87], thus showing promise. However, its long-term effects need to be tested in further studies.

#### 4-Hydroxy Isoleucine

One of the most studied fenugreek compounds is 4-Hydroxyisoleucine, (4-HIL), classified as an amino acid that constitutes 1–3% of fenugreek seeds and 0.5–1.5% of the leaves. It has been shown to have antidiabetic effects in sucrose-lipid-fed diabetic rat models, regulating pancreatic insulin secretion, stimulating insulin secretion in vivo and improving glucose tolerance [70]. It has been proposed that 4-HIL works by negatively regulating TNF-alpha production, resulting in insulin sensitivity and increasing GLUT4 (glucose transporter type 4) and p-IRS-1 (insulin receptor substrate 1) in the insulin-signaling pathway [88]. It is thought to work directly on pancreatic cells and may enhance glucose uptake in peripheral tissues such as skeletal muscle and adipose tissue, contributing to improved glycemic control.

Oxidative stress and inflammation play crucial roles in the pathogenesis of T2DM and its complications. 4-HIL exerts its antioxidant and anti-inflammatory effects by decreasing oxidative stress and the inflammatory response via activation of nuclear factor erythroid 2-related factor 2 (Nrf2 gene, a free radical scavenger and antioxidant enzyme stimulator) and the transforming growth factor β1 signaling pathway [71].

### 2.5. Fenugreek and the Microbiome

The role of the microbiome in maintaining health is well established, and the tools for examining the microbiome and the consequences of the metabolic activity of the microbiome, are readily available. Here, we review studies that consider the interaction of fenugreek formulations on the microbiome.

There have been a limited number of studies to date, and these have mostly utilized mouse or rat models. In one series of studies, young male C57BL/6J mice were fed control or high-fat diets with or without 2% fenugreek seed powder over 14–16 weeks. The first study indicated improved glycemic control and improved lipid profile in mice fed with a high-fat diet, including fenugreek [59]. A follow-up study with the same model analyzed changes in the microbiome and found that the inclusion of fenugreek could reverse changes in the microbiome induced by a high-fat diet [89]. Fenugreek restored the distribution of microbial taxa to one associated with a good metabolic profile. The inclusion of fenugreek in the diet did not change the variety of taxa, but only the relative proportions of species. Further studies with the same model [90] confirmed an increase in the proportion of the *Verrumicrobia phylum* in mice fed fenugreek seed powder due to an increase in the genus *Akkermansia*, which includes the mucin-degrading species *Akkermansia muciniphilia*. Decreased proportions of *A.muciniphilia* in the microbiome have been associated with adverse metabolic conditions, including obesity and diabetes, in humans and animals [91].

Other studies have examined the effect of fenugreek seed components on the microbiome. In a study on peripheral neuropathy caused as a side effect of oxaliplatin chemotherapy [92], diosgenin could relieve the symptoms of peripheral neuropathy. Diosgenin restored the variety of the microbiome in mice treated with oxaliplatin. Importantly, fecal material transferred from diosgenin-treated mice could also relieve the symptoms of peripheral neuropathy, suggesting a direct therapeutic effect of diosgenin mediated through the microbiome. In a model of osteoporosis using ovariectomized rats, treatment with diosgenin could improve bone loss [93]. The osteoporotic rats suffered a loss of microbiome composition, which was restored after treatment with diosgenin.

Trigonelline was found to inhibit the conversion of choline to the pro-atherosclerotic metabolite trimethylamine-N-oxide (TMAO) in isolated cultures of anaerobic microbes isolated from human feces [94]. Finally, the polysaccharide galactomannan, the major component of fenugreek seeds, was found to improve glycemic control in 7–8-week-old C57 BL/6 J mice, which was associated with a healthier microbiome [95].

Taken together, these studies suggest a strong, protective effect of fenugreek seed formulations on the microbiome, either as a complete formulation or as individual components.

### 2.6. Mechanisms Involved in the Antidiabetic Effects of Fenugreek

The mechanisms involved in the antidiabetic properties of fenugreek have been investigated [57,58,60,61,62,63,64,75,96,97,98], but more research is needed to identify all of the involved biomolecules and pathways involved. Fenugreek seeds are thought to exhibit their antidiabetic effects in a number of ways, including stimulation of insulin release from beta cells and improved insulin signaling [61], glucagon suppression and delaying gastric emptying [98]. An in vitro study using human hepatoma cells (HepG2) showed that under normal and hyperglycemic conditions, fenugreek seed extract significantly enhanced glucose uptake in comparison to metformin and insulin [58]. Additionally, fenugreek seeds have demonstrated an ability to improve insulin sensitivity in adipocytes [57], which is crucial for regulating blood sugar levels and mitigating the risk of excessive insulin secretion. Fenugreek seeds have been shown to inhibit intestinal sodium-dependent glucose uptake in vitro using rabbit intestinal brush border membrane vesicles [63]. Fenugreek seeds have also been shown to regulate glucagon-like peptide-1 (GLP-1) activity, which is important for secreting insulin after food intake [97].

Fenugreek extracts have also been shown to control blood glucose levels by targeting *α*-amylase and *α*-glucosidase enzymes, which are required for metabolizing complex carbohydrates [89]. Fenugreek exerts hypolipidemic effects [28,62,99], hence targeting molecules such as triglycerides, which are thought to contribute to its antidiabetic effects since alterations in lipoprotein levels have been associated with an increased risk of diabetes [100]. Membrane fluidity, which is impaired in those with diabetes, has been shown to improve with fenugreek, thus improving glucose transport and vesicular trafficking [60]. In addition, the antioxidant properties of fenugreek may also be instrumental in reducing oxidative stress damage that has been linked to diabetes [71,101]. Several studies have investigated or proposed the antidiabetic mechanisms of fenugreek through in vitro and in vivo animal studies [52,57,58,60,61,62,64,65,66,67,68,69,70,75,101] that we have summarized in Figure 1. Understanding the contribution of the individual bioactive ingredients found in fenugreek is needed to learn how they exert their antidiabetic effects.

## 3. Discussion

Fenugreek (Trigonella foenum-graecum) has received considerable scientific interest for its potential therapeutic applications, particularly in the management of diabetes mellitus. This attention is due to its rich composition of bioactive compounds, including 4-hydroxyisoleucine (4-HIL) and trigonelline, among others [86]. Research has shown that fenugreek exerts hypoglycemic effects through various mechanisms (Figure 1), such as enhancing insulin sensitivity and stimulating insulin secretion. Additionally, fenugreek exhibits antioxidant properties, which can mitigate oxidative stress and inflammation commonly associated with diabetes and its complications [102]. Fenugreek has demonstrated potential in managing lipid profiles by reducing total cholesterol, LDL cholesterol and triglycerides, addressing another critical aspect of diabetes management [27,28]. Research into the antidiabetic mechanisms of individual fenugreek components has revealed their biological targets and some of the pathways involved. Diosgenin may regenerate pancreatic beta cells and insulin granules [66]. Trigonelline increases insulin sensitivity and improves insulin signaling pathways [67,68]. Galactomannan can improve lipid profiles [69] and glycemic control [95], and 4-HIL stimulates insulin secretion [69,88] and has anti-oxidative and anti-inflammatory effects [71]. Many conventional diabetes medications, such as Metformin, primarily focus on a single target, such as insulin sensitivity or hepatic glucose production [103]. However, since fenugreek and its bioactive components have multiple biological targets and impact various biochemical pathways (such as enhancing insulin secretion and sensitivity, modulating glucose absorption and enzyme activity, providing anti-inflammatory and antioxidant effects, and improving lipid metabolism), it may offer enhanced efficacy in managing T2DM.

Additionally, in comparison to prescribed synthetic drugs, being of natural origin and having been consumed through diet for many generations with noted health benefits [51], natural products are likely to have a lower risk of unwanted side effects. Animal studies using controlled administration of doses (up to 1000 milligrams of fenugreek seed per kilogram of weight) have been shown to be non-toxic to mammals [65]. Bioactive components of fenugreek have also shown promise in regard to safety. Doses of up to 16 g of a galactomannan derivative given to human volunteers showed no side effects whilst being able to reduce glucose levels after eating [87]. However, this is an area that needs further investigation since it is known that many plants produce toxic compounds as protection mechanisms [104], so purification methods need to be optimized to ensure any toxic components are removed. The precise mode of action of fenugreek and its active components need to be fully elucidated prior to administering it as a drug to understand any potential adverse effects it may have. Several promising clinical trials with fenugreek seeds and isolated bioactive compounds with T2DM patients have been observed [26,27,76,77,78,79,80,81,86,87,105,106]; however, heterogeneity in the participant’s diabetes status, low participant numbers and the wide ranges of doses used and the duration of treatment given warrant further studies.

### 3.1. Bioavailability and the Microbiome

One limitation of using natural products as therapeutics is their bioavailability/bioaccessibility. This refers to the amount available for absorption in the intestines, remaining soluble and stable before cell absorption and thus being able to exert its bioactive effects. Poor intestinal absorption is one reason why several promising bioactive natural products, such as curcumin, have not yet developed into therapeutics [107]. Unlike curcumin, galactomannan isolated from fenugreek is very bioaccessible [108]. When used in combination with other bioactive compounds, galactomannan has been used as a drug-delivery molecule to enhance the bioavailability of other bioactive compounds such as curcumin as observed in experiments with rat models [109] and studies involving human participants [110].

Understanding the bioavailability of fenugreek and its interaction with the microbiome is crucial in exploiting fenugreek as a preventative and/or diabetes treatment. Limited studies have shown that fenugreek seeds have a protective effect on the microbiome [92]. The gut microbiome can influence the efficacy of drugs (and vice versa) in an individual [111] and thus influence its bioavailability and toxicity. More experiments looking at the interaction of fenugreek and its bioactive components are urgently needed to make effective therapeutics in the future.

### 3.2. Limitations and Future Directions

The heterogeneity and variability of clinical trial results are related to several factors, namely the diabetic state of the participant (healthy versus diabetic patients), other conditions they may have, their age, the duration of treatment, the administered dosages, and the fenugreek preparation methods used. Moving forward, having standardized methods for the preparation and isolation of bioactive compounds and being able to produce these in large quantities will facilitate testing and further clinical trials. 4-HIL has shown promise as a potential diabetes treatment; however, current methods for producing it in *Escherichia coli* are not economically efficient [112], and hence better, more reproducible methods are needed. Further studies also need to ascertain whether fenugreek and its active compounds interact with other prescribed medications since many people with T2DM take medications for other conditions in addition to their T2DM. One study reported that fenugreek interfered with the action of warfarin in a patient who was prescribed it for atrial fibrillation [113]. Investigating dose response is important for ascertaining safe doses for prescription. Combining fenugreek with conventional therapies such as metformin, sulfonylureas, or insulin may lead to improved glycemic control and reduced medication dosages, thereby minimizing adverse effects and enhancing patient adherence to treatment regimens. Trials combining prescribed medications such as metformin and natural products such as fenugreek have shown decreased fasting blood glucose, HbA1c levels and cholesterol when used in combination and compared to the placebo group [105].

More comprehensive clinical trials are needed to evaluate the long-term effects of fenugreek supplementation on glycemic control, insulin sensitivity and diabetes-related complications in diverse populations, including individuals with type 1 and type 2 diabetes, prediabetes and gestational diabetes, an under-researched area. More under-represented groups should be included in clinical trials since T2DM disproportionately affects non-white ethnic populations, and hence, more participants from non-white backgrounds should be included in future trials [114].

More research is also needed into the preventative effects of fenugreek. In addition to being used by those with T2DM, being able to prevent T2DM will prevent damage to the body and the onset of diabetes complications. A study by Gaddam et al. [115] showed that participants with prediabetes who were given 10 g a day over a 3-year period were four times less likely to develop T2DM compared to controls. This long-term study highlights the potential of fenugreek as a preventive measure for T2DM. Similarly, Neelakantan et al. [77] and Kim et al. [80] conducted a meta-analysis of clinical trials and found that fenugreek consistently reduced fasting blood glucose and HbA1c levels among T2DM patients, reinforcing the beneficial effects of fenugreek in glycemic control. Another study by Hadi et al. [106] revealed substantial decreases in fasting plasma glucose levels among T2DM patients supplemented with fenugreek seed powder, further substantiating its potential to enhance glycemic control. These studies collectively underscore the importance of fenugreek in the management of T2DM, suggesting that its bioactive compounds may improve insulin secretion and sensitivity, modulate glucose absorption and enzyme activity, and provide anti-inflammatory and antioxidant effects, which all contribute to better glycemic control.

The molecular mechanisms behind the mode of action of fenugreek in T2DM prevention and management are slowly being elucidated (Figure 1). The antidiabetic effects are largely due to fenugreek’s hypoglycemic effects, reducing blood glucose levels by targeting various molecules, receptors and membranes. Fenugreek also appears to decrease oxidative stress and lipoprotein levels, which also contribute to its antidiabetic effects. More mechanistic studies are required to better understand the underlying pathways through which fenugreek and its bioactive compounds affect glucose metabolism and insulin sensitivity. Technical advances being made in molecular docking [116] and artificial intelligence [117] will no doubt help advance our knowledge and understanding of fenugreek’s potential as an antidiabetic therapy by complementing current methods.

## Figures and Tables

**Figure 1 ijms-25-06987-f001:**
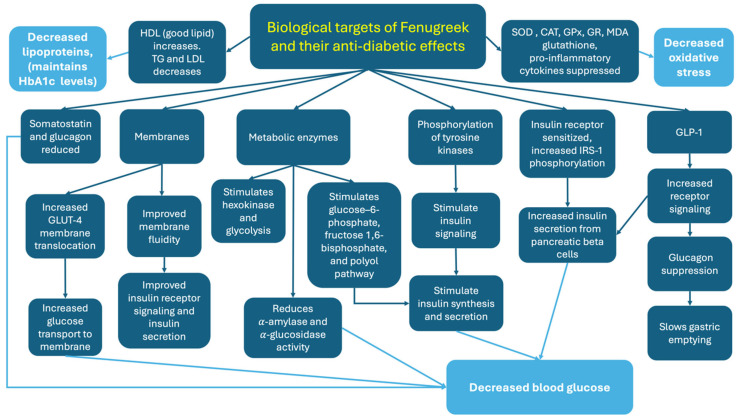
Proposed biological targets of fenugreek and their molecular antidiabetic effects summarized from reviewing in vitro experiments and in vivo animal studies. Abbreviations: HbA1c—hemoglobin A1c, HDL—high-density lipoprotein, TG—triglycerides, LDL—low-density lipoproteins, SOD—superoxide dismutase, CAT—catalase, GPx—glutathione peroxidase, GR—glutathione reductase, MDA—malondialdehyde, GLUT-4—glucose transporter type 4, IRS-1—insulin receptor substrate-1, GLP-1—glucagon-like peptide-1.

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
