# Peer review of "The Role of Fenugreek in the Management of Type 2 Diabetes"

_ijms, 2024, doi:10.3390/ijms25136987_

Round 1

Reviewer 1 Report

Comments and Suggestions for Authors

The manuscript entitled "The role of Fenugreek for the management of type 2 diabetes" examines of Fenugreek's potential in diabetes care. 

The manuscript makes a substantial contribution by detailing the biochemical mechanisms and therapeutic potential of Fenugreek in managing type 2 diabetes. It incorporates a broad array of studies, from molecular to clinical, providing a robust discussion on Fenugreek's efficacy and safety. The focus on a naturally derived compound with fewer side effects compared to conventional diabetes medication is particularly relevant given the rising interest in alternative and complementary medicine.

Some sections would benefit from further clarification and detail for readers not familiar with the biological and chemical terminology. For instance, the description of insulin signaling could include simplified diagrams to aid understanding.

There are a few instances where terminology and definitions vary; standardizing these would improve readability and professional quality.

Some claims provided in the manuscript might be perceived as slightly optimistic regarding the efficacy of Fenugreek without sufficient emphasis on the mixed results and limitations of some studies.

The manuscript would benefit from a more detailed discussion of the limitations of the current studies on fenugreek, including the heterogeneity in study designs and sample sizes, which may affect the generalizability of the results.

Comments on the Quality of English Language

Minor typographical and grammatical corrections could enhance the clarity of the text.

Author Response

Thank you for taking the time to read our manuscript and provide very useful comments and suggestions. We agree with all of your comments and have addressed each one below and updated our manuscript accordingly (uploaded). We believe these changes have now greatly improved the quality and content of our manuscript.

We have also made some additional changes based upon suggestions made by reviewer 2 to remove much of the non-essential text about diabetes and focus more on fenugreek, which we have now done. In the attached revised manuscript, we have now deleted unnecessary text and new text, or text moved from elsewhere is highlighted in red. We have also added new references (highlighted in red in the reference list) to reflect these changes.  

Point 1: Some sections would benefit from further clarification and detail for readers not familiar with the biological and chemical terminology. For instance, the description of insulin signalling could include simplified diagrams to aid understanding.

Thank you, we agree that some readers may be unfamiliar with the terminology so we have added some additional text for clarity for examples, we have defined prediabetes (lines 35-36),  described phenolic compounds (lines 182-183), and defined HbA1c (line 263). We have also added more methodological details used in clinical trials for clarity (rodent models used and doses of fenugreek administered) for clarity.

The introduction of a new figure is a great idea. We have introduced a new, more appropriate figure (now figure 1). This shows in a graphical format how fenugreek is exerting its anti-diabetic effects. Here we also include how this links in with insulin signalling as you have requested. The previous figure 1 table 1 have now been deleted as per the other reviewer’s advice.

Point 2: There are a few instances where terminology and definitions vary; standardizing these would improve readability and professional quality.

Thank you, we agree with your comments and agree this may cause confusion and needs to be corrected. We have gone through the manuscript and paid careful attention to scientific terminology and definitions and tried to standardise for clarity. Examples include therapeutics (instead of drugs, pharmaceuticals), compound (instead of component, product), T2DM (instead of type 2 diabetes, late-onset diabetes), beta cells instead of β-cells.

Point 3: Some claims provided in the manuscript might be perceived as slightly optimistic regarding the efficacy of Fenugreek without sufficient emphasis on the mixed results and limitations of some studies. The manuscript would benefit from a more detailed discussion of the limitations of the current studies on fenugreek, including the heterogeneity in study designs and sample sizes, which may affect the generalizability of the results.

Thank you, we agree with your comments and with hindsight, we should have been more critical of previously published studies. We have now updated several sections of our manuscript to highlight the limitations of these studies (e.g. lines 247, 264, 318) and have rearranged the discussion so there is now a whole section entitled “ Limitations and future directions” to address this point in more detail (from lines 475).

Point 4: Minor typographical and grammatical corrections could enhance the clarity of the text.

Thank you. We agree that the quality of the written language and attention to detail needs to be improved. We have now gone through the whole manuscript in detail and paid particular attention to spelling and grammar and made some minor adjustments to improve the overall quality of writing throughout, including checking nomenclature and scientific terminology to improve clarity. We think that the overall quality and content of our manuscript has been improved by incorporating your suggestions and feedback, thank you.

Reviewer 2 Report

Comments and Suggestions for Authors

The topic of this article is the role of fenugreek in the management of type 2 diabetes, but the introduction length of fenugreek is only from line 177 to line 299. There is too much content describing the influencing factors and treatment measures of diabetes, and the overall focus is not enough.

  The use of many natural products in diabetes management is listed in Table 1, but this deviates from the subject of this article and more attention should be paid to fenugreek itself. Moreover, the content in Table 1 has nothing to do with the topic of the entire paper and cannot highlight the main points of the paper.

Fenugreek is not mentioned anywhere in the introduction, so the framing of the article is problematic.

  When introducing the hypoglycemic effect of fenugreek, the author should introduce the mechanism of action in depth instead of simply listing the experimental results as now.

Comments on the Quality of English Language

Needs further improvement.

Author Response

Thank you for taking the time to read our manuscript and provide very useful comments and suggestions. We agree with all of your comments and have addressed each one below and updated our manuscript accordingly. We believe these changes have now greatly improved the quality and content of our manuscript.

We have also made some additional changes based upon suggestions made by reviewer 1 to clarify some terminology and to emphasise the limitations of previous studies. We have highlighted them also in the uploaded revised manuscript. Some text has been deleted and new text, or text moved from elsewhere is highlighted in red. New references required to support our manuscript are highlighted in red in the reference list.

Point 1: The topic of this article is the role of fenugreek in the management of type 2 diabetes, but the introduction length of fenugreek is only from line 177 to line 299. There is too much content describing the influencing factors and treatment measures of diabetes, and the overall focus is not enough.

With hindsight, we agree that much of content in the introduction is unecessary and digresses from the main focus of the paper (Fenugreek). We have now deleted a considerable amount of text in the sections “Introduction to Diabetes Mellitus” including the section “Risk Factors for Type 2 Diabetes mellitus” and “Current methods for the management of type 2 diabetes”. Deleted text has been highlighted with a line through it.

Point 2: The use of many natural products in diabetes management is listed in Table 1, but this deviates from the subject of this article and more attention should be paid to fenugreek itself. Moreover, the content in Table 1 has nothing to do with the topic of the entire paper and cannot highlight the main points of the paper.

Thank you, we agree that the introduction contains unnecessary text that digresses from the main focus of the paper (Fenugreek) so we have deleted Figure 1 and Table 1 from our manuscript as per your advice.   

Point 3: Fenugreek is not mentioned anywhere in the introduction, so the framing of the article is problematic.

Thank you, we fully agree. This is an oversight and we have now rectified this. Before the “Main text” section we have now introduced a section about Fenugreek at the end of the introduction. The new text can be found in lines 142-158. This has been moved from lines 198-205 and 184-193 in the originally submitted text.

Point 4: When introducing the hypoglycemic effect of fenugreek, the author should introduce the mechanism of action in depth instead of simply listing the experimental results as now.

Thank you. We agree. For clarity we decided to introduce a new figure (to replace the deleted Figure 1) to show the mechanistic hypoglycaemic and anti-diabetic effects of fenugreek. This is now labelled Figure 1 and is an original figure that we have designed based on previous evidence from the scientific literature. We have also introduced some text to introduce the new figure, that can be found in lines 382-401 and is summarised in the discussion also 424-429. We have also included additional text  summarising what is known about the bioavailability of fenugreek in lines 456-466 as this is directly related to its bioactive effects and potential as a therapeutic.

Quality of English Language - Needs further improvement.

Thank you. We agree that the quality of the written language could be improved. We have now gone through the whole manuscript in detail and paid particular attention to spelling and grammar and made some minor adjustments to improve the overall quality of writing throughout, including checking nomenclature and scientific terminology.  We think that the overall quality and content of our manuscript has been improved by incorporating your suggestions and feedback, thank you.

Round 2

Reviewer 2 Report

Comments and Suggestions for Authors

The level of the manuscript was not significantly improved.

Comments on the Quality of English Language

The level of the manuscript was not significantly improved.

Author Response

Dear reviewer,

Thank you for your feedback. We have again, improved our manuscript. Attached is our 'clean' version with your changes incorporated. 

I hope you agree that this version is greatly improved.

Many thanks

Dr Terry 
